# Weekly and Daily Tooth Brushing by Care Staff Reduces Gingivitis and Calculus in Racing Greyhounds

**DOI:** 10.3390/ani11071869

**Published:** 2021-06-23

**Authors:** Nicola J. Rooney, Katharine L. Wonham, Katherine S. McIndoe, Rachel A. Casey, Emily J. Blackwell, William J. Browne

**Affiliations:** 1School of Veterinary Science, University of Bristol, Langford BS29 6BN, UK; Katie.wonham@thedonkeysanctuary.org.uk (K.L.W.); km15575@bristol.ac.uk (K.S.M.); emily.blackwell@bristol.ac.uk (E.J.B.); 2Donkey Sanctuary, Sidmouth EX10 0NU, UK; 3Dog’s Trust, London UB9 6JW, UK; Rachel.Casey@dogstrust.org.uk; 4School of Education, University of Bristol, Bristol BS8 1JA, UK; william.browne@bristol.ac.uk

**Keywords:** teeth, greyhound, intervention, brushing, calculus, gingivitis, dental, periodontal

## Abstract

**Simple Summary:**

Dental disease affects many dogs worldwide and is believed to be particularly problematic for racing greyhounds. It costs the industry and rehoming charities financially and likely causes unnecessary suffering to a large number of dogs. The risk factors for dental disease in this population are debated, and the best methods to overcome it are relatively unresearched. We carried out a trial in which 160 racing greyhounds were divided into three groups. The staff looking after them either brushed their teeth daily, weekly or never, for a period of two months. An experimenter measured the dog’s level of calculus (hardened dental plaque) and gingivitis (gum inflammation) at the start, and again after two months. We found that both weekly and daily brushing resulted in significant reductions in calculus, but for gingivitis only daily brushing resulted in a significant reduction. The effects, however, were not noticeable on the front incisor teeth. Since the staff implementing the routine reported a minimal time commitment and positive experiences, we suggest that daily brushing is recommended for racing greyhounds, and that emphasis is placed on brushing all teeth groups. Similar trials need to be conducted with retired greyhounds since these have been shown to present particularly high levels of periodontal disease.

**Abstract:**

Periodontal disease is one of the most common conditions affecting dogs worldwide and is reported to be particularly prevalent in racing greyhounds. A range of potential risk factors have been hypothesised. Previous research has suggested that regular tooth brushing can reduce both calculus and gingivitis, but the frequency required is unclear. Here, we report a controlled blinded in situ trial, in which kennel staff brushed 160 racing greyhounds’ teeth (living at six kennel establishments), either weekly, daily or never over a two-month period. All of the visible teeth were scored for calculus and gingivitis, using previously validated scales. We calculated average scores for each of the three teeth groups and overall whole mouth scores, averaging the teeth groups. Changes were compared to the baseline. After two months, the total calculus scores (controlling for baseline) were significantly different in the three treatment groups, (F_(2,129)_ = 10.76, *p* < 0.001) with both weekly and daily brushing resulting in significant reductions. Gingivitis was also significantly different between groups (F_(2,128)_ = 4.57, *p* = 0.012), but in this case, only daily brushing resulted in a significant reduction. Although the dogs in different kennels varied significantly in their levels of both calculus (F_(5,129)_ = 8.64, *p* < 0.001) and gingivitis (F_(5,128)_ = 3.51 *p* = 0.005), the intervention was similarly effective in all of the establishments. The teeth groups varied, and the incisors were not significantly affected by the treatment. Since the trainers implementing the routine, reported a minimal time commitment and positive experiences, we suggest that daily brushing is recommended for racing greyhounds, and that any instructions or demonstrations should include attention to all teeth groups including the incisors. Similar trials need to be conducted with retired greyhounds since these have been shown to present particularly high levels of periodontal disease.

## 1. Introduction

There are 15,000 active racing greyhounds in the UK alone [1], with likely four times as many retired dogs, and commercial racing occurs in countries across four different continents. The welfare of racing greyhounds is frequently under the spotlight, and whilst the majority of the attention features on injuries during racing, as the number of dogs successfully retiring and being rehomed increases, there is growing pressure to consider the dogs’ welfare for the entirety of their lives. There are a number of late onset health issues that have started to manifest as the proportion of ex-racers reaching old age increases. One issue deemed to be of significant welfare concern for the greyhound breed [2] and also one of the most ubiquitous health concerns for companion dogs in general, is periodontal disease [3]. 

Greyhounds are generally believed to be prone to periodontal disease [4], and although comparative age-controlled studies are lacking, primary veterinary clinic records suggest that it is a very commonly presented issue (39% of dogs; [5]). Dental health is an important financial issue when rehoming racing greyhounds, as the Retired Greyhound Trust reported that 14% of their funds goes on dental treatment [1]. Periodontal disease can significantly compromise welfare, affecting the dog’s ability to eat and behave normally as well as causing pain and discomfort [6], and exacerbating other serious systemic conditions [7,8]. 

The reason for this apparent breed disposition is debated. Some implicate the long skull shape, whilst others assert that causes may be the non-solid diet that is commonly fed in racing kennels (e.g., see [9]), or the lack of dental care during early rearing and racing careers [10]. An intervention suitable for improving dental health in dogs within an industry that is time poor and financially restricted is therefore needed. 

A major aspect of periodontal disease is plaque, a biofilm layer that forms on the tooth within hours of prophylaxis, within which bacteria colonise [11]. If plaque is not removed, it mineralises, forming calculus, composed primarily of calcium carbonate deposited between bacterial remnants within twelve days [12]. Calculus is inert and does not stimulate an immune response, but provides a porous surface for plaque to build on and facilitates the growth of anaerobic and more pathogenic bacteria [12,13]. Studies have shown that prophylaxis in the form of tooth brushing removes plaque from the buccal surfaces of the tooth and therefore can prevent calculus build up [12], but many believe that once it matures, calculus can only be removed by dental scaling [14]. 

A further dimension of periodontal disease is gingivitis; inflammation of the gum, swelling, oedema and bleeding [15]. Gingivitis has been demonstrated to be reversible, if the provoking factors are removed and the tissues recover [16] and has been shown to be reduced by regular tooth brushing. Periodontitis describes the inflammation and progressive destruction of the periodontal ligament and the alveolar bone. This is irreversible and can eventually lead to the loss of the tooth [16].

When developing methods to score periodontal health in racing greyhounds, our research [17] indicated that the progression of periodontal disease is not always linear, as some previous studies have assumed. Many authors have utilised a single periodontal disease scale that simultaneously takes into account levels of gingivitis, calculus and gum recession, and assumes their development is progressive (e.g., [18]). Our own studies, however, saw that some greyhounds had high levels of gingivitis, yet little calculus and vice versa [17]. In a point sample study of almost 500 racing and retired greyhounds [19], we saw high levels of periodontal disease. We identified several risk factors associated with between-dog variation including, for example, feeding leftover human food. We identified different risk factors for each aspect of periodontal disease, which suggests that each needs to be considered separately. We also saw that reported regular tooth brushing was associated with reduced levels of both calculus and gingivitis, although the frequency required to be effective was unclear. Buckley et al. [20], similarly found that in a mixed breed population of 17,184 dogs, those reported to have their teeth brushed (or provided dental chews) daily generally had better oral health. 

In other breeds, predominantly beagles, there have been controlled trials examining the efficacy of brushing. Several have shown that brushing can be effective at mitigating plaque build-up and, consequently, gingivitis (e.g., [11]), whilst the addition of a clinically proven product can increase the efficacy of brushing [21]. A laboratory-based trial suggested that three times per week is the critical frequency to prevent gingival inflammation [22]. However, it could be argued that it is unlikely that owners or trainers will remember to brush three times a week and a simpler direction of once a week or once a day, would more likely be followed. In fact, a follow-up study using sites of experimental gingivitis showed that, only by brushing every day, could clinically healthy gingivae be obtained [23]. Harvey et al. [24] similarly highlighted every other day to be the critical frequency, whilst Ingram and Gorrel [25] saw that daily brushing was required to reduce gingivitis in two-year old Labrador Retrievers. In contrast, an experimental trial suggested tooth-brushing every other day did not maintain clinically healthy gingivae in dogs, but the addition of dental chews, every other day, decreased gingivitis scores and reduced the accumulation of dental deposits [26]. 

Past research has generally occurred in a clinical setting and involved anesthetising dogs to scale and polish their teeth. The trials were started with a clean mouth and the effectiveness of brushing by a trained technician at reducing plaque and calculus accumulation were examined. However, it remains unclear whether brushing is an effective strategy to reduce aspects of periodontal disease for dogs who already have plaque and calculus build-up, as is often the case with older dogs and racing greyhounds. We are unaware of any previous trials evaluating the impact of brushing on greyhounds with variable existing periodontal health. This study is also unique in using lay rather than trained operatives to carry out brushing.

Here, we describe a blinded controlled in situ trial, aimed at ascertaining the effectiveness of kennel staff brushing racing greyhounds’ teeth either weekly or daily with an enzymatic toothpaste. We used oral examination of conscious dogs and, hence, neither probing nor staining were appropriate for scoring dental health. Instead, we used scales that have been previously developed and validated for use on racing greyhounds [17]. Since any intervention needs to be quick and practical for time-poor trainers to implement, we surveyed trainers about the perceived value and time costs of brushing. To ensure interventions were welfare compatible, we also collected data on the dogs’ behaviour during oral manipulation pre- and post-trial. 

## 2. Materials and Methods

### 2.1. Subject Recruitment 

We contacted approximately 20 greyhound trainers (individuals licensed to race dogs and contracted to provide accommodation, care and conditioning for racing dogs), regularly racing greyhounds at one of two Greyhound Board of Great Britain (GBGB) tracks in the Southwest of England. Inclusion criteria were that the trainer must have at least twelve current racing dogs and have reported in a trackside survey [19], not to currently have a regular programme of teeth brushing. Trainers were initially contacted by telephone and the study design and time commitment were explained. The incentive of GBP 50 worth of dog health supplies was also offered. 

A total of six trainers volunteered to participate. They were called a second time to allow them to ask further questions and, if they were still happy to participate, a visit was arranged. Each trainer was visited by two researchers (KW and NJR) at their racing kennel on three occasions, over a two-month period. At the time of the first visit, all of the healthy actively racing or retired dogs present at the kennel establishment were recruited onto the trial. Any dog deemed too fearful, or that was likely to be rehomed in the next two months was excluded, leaving a total of 168 greyhounds. None of the participating dogs had any dental treatment or therapeutic intervention (e.g., given antibiotics) likely to affect their teeth during the course of the study. Since some trainers designated other care staff to conduct the trial, we refer to “trainers” as the participating individual hereafter. 

### 2.2. Teeth Scoring 

On each visit, every greyhound’s teeth were scored for calculus and gingivitis, using a previously validated scale [17]. On the first visit, the dogs were each scored by the second author (KW), a qualified veterinarian. The trainer or the experimenter (NJR) brought the dog to the scorer. The scorer then examined the left-hand side of the mouth. Only the left-hand side was examined, as this was previously shown to save considerable time, and to correlate highly (0.98) with the whole mouth score [17]. The amount of calculus on each visible tooth was scored; this excluded the molars, which required too much jaw manipulation for examination in a conscious dog, and the first pre-molar on the lower jaw (P1), which was often covered by the tongue. The scale used scored teeth from 0–3, was modified from Greene and Vermillion [27] and validated in greyhounds ([17]; Table 1). 

We also rated the level of gingivitis of the gum adjacent to each tooth, using an index of scores from 0–3, modified from Kyllar and Witter [28] and similarly validated ([17] Table 2).

### 2.3. Group Allocation

The dogs within each establishment were then divided into the following three groups: brush daily, brush weekly and control. Effort was made to balance the groups for initial dental scores and for age; however, for dogs that were pair-housed, those in the same individual kennel unit were allocated to the same group to facilitate ease of treatment by care staff. Colour coded signs and a record sheet were placed on the kennel door, on which the care staff recorded each time the dog’s teeth were brushed. 

### 2.4. Brushing Protocol 

The second experimenter demonstrated to the care staff (either trainer or their employee(s)), how to brush the dogs’ teeth. The aim was to brush for approximately 45 s, to cover all teeth groups, on both sides of the mouth, using a soft circular motion on each tooth surface. The trainers were encouraged to be gentle with the dogs and to reward them for calm behaviour. If a dog were to show behavioural signs of fear during brushing, they were advised to discontinue, to contact the research team, and the dog would be removed from the study. This never occurred. 

Each carer was given several tubes of toothpaste. They were in plain tubes labelled daily and weekly, and the trainers were unaware that all tubes were the same, containing an enzyme complex (Amylase, Glucose Oxidase, Potassium Thiocyanatem Kactoferrine, Lactoperoxydase, Iysozyme, Superoxide, Dismutase). Establishments were given written instructions for staff who were not present at the time, reiterating what was communicated in the demonstration. Trainers and care staff were urged not to change the dogs’ routines, nor to treat the treatment groups differently in any other way. For example, if establishments usually gave bones, they were asked to continue this practice for all dogs. Weekly phone calls and an interim follow-up visit to each establishment aimed to increase compliance. 

### 2.5. Follow Up Visits 

After 28 (±2) and 56 (±2) days, each dog was re-examined by the same experimenter. Due to dogs leaving the establishment, and trainers accidentally swapping treatment groups, eight dogs were lost by visit 3, giving a retention rate of 95% and 160 dogs completed the two-month trial. The final sample included three retired, and two pre-racing dogs. The dogs were aged between 5 and 111 months (mean = 37.3 ± 15.31 months). There were 78 males and 82 females; and the number of dogs per trainer ranged from 18 to 37 (Table 3). The timing of the visits was arranged to ensure that no dog had already had its teeth brushed on the day of the visit, as this could exacerbate gingivitis.

On each follow-up visit, the scorer was positioned behind a screen and the second experimenter facilitated the dogs being brought individually in a random order. Teeth scores were recorded blindly against dog ID numbers and were only subsequently linked to the dog’s name and treatment group by the second experimenter. The trainer or carer responsible for each dog was also interviewed to collect information on any changes to routine, illness or injury. 

In addition, on visit three, trainers were asked about their feeding and bone provision routines, whether they thought each individual dog’s dental health had improved, deteriorated or stayed the same over the past two months. They were also asked general questions about how long the brushing routine took, whether levels of calculus and gum inflammation had changed with either daily or weekly regimes, how the dogs’ reactions to brushing had changed and their perception of the value of the intervention (Table 3). Interviews were conducted by NJR; therefore, KW remained blind to group allocation. The trainers were given GBP 50 worth of veterinary products as compensation for their time. 

### 2.6. Calculation of Dental Scores

Average scores per tooth group were calculated for the incisors (*n* = 6), canines (*n* = 2) and pre-molars (*n* = 7) on the left-hand side of the mouth for both calculus and gingivitis. For calculus, averages were based only on those teeth present. The scores were also used to calculate a whole mouth score (both for calculus and gingivitis) by averaging the three groups, whilst eliminating any missing teeth for calculus scores (whole mouth score = (incisors + canines + PM)/3). Each tooth group was analysed separately, as well as the whole mouth scores (total of eight measures), to explore effects of brushing on specific mouth regions. 

### 2.7. Analysis

We compared the baseline measures between the three groups using a one-way ANCOVA. We then carried out linear modelling to explore the effect of baseline score, treatment group and kennel establishment on each of the eight variables (whole mouth, incisor, canine and pre-molar average) for both calculus and gingivitis. The results for the model that includes all significant predictors are displayed (Table 4), therefore, for example, if the effect of kennel was non-significant, the model with just the treatment group was used.

## 3. Results

### 3.1. Dental Scoring

At the outset, the average level of calculus in the whole sample over the whole mouth was 0.95 (±SD 0.503) and that of gingivitis was 1.08 (±0.823). There were no significant pre-treatment differences between the three groups for any of the eight measures of dental health (*p* > 0.05 for all; Table 4).

Since visit two was designed mainly to increase compliance and initial analysis (on whole mouth scores) suggested that the results were cumulative (Figure 1), subsequent analysis used only the baseline and visit three data.

When testing the effect of a dog’s baseline score (scores on visit 1), treatment group and kennel establishment on each of the eight dental variables, the baseline score was significant for all of the variables (Table 4). Dogs in different kennels differed significantly in their levels of both calculus (F_(5,129)_ = 8.64 *p* < 0.001) and gingivitis (F_(5,128)_ = 4.57 *p* = 0.005) (after adjusting for baseline) in the whole mouth, and all individual teeth subgroups except the incisors. 

After two months, the whole mouth calculus scores were significantly different in the three groups (F_(2,129)_ = 10.76 *p* < 0.001; Table 4), with both weekly and daily brushing resulting in significant reductions (*p* < 0.001; Table 5). Gingivitis was also reduced, but in this case, only daily brushing resulted in a significant change (*p* = 0.03; Figure 2). 

When examining the individual tooth groups, we found that the calculus levels on the incisors were not significantly affected by the treatment, whilst canine and pre-molar calculus were significantly reduced by both weekly and daily brushing (Table 5). Similarly, for gingivitis, the incisors displayed no significant treatment effect, whilst canine and pre-molar gingivitis were significantly reduced by both weekly and daily brushing (Table 5). In general, the intervention was similarly effective at all of the establishments, but there was only a significant interaction between the kennel and the treatment group for pre-molar gingivitis (F_(10,119)_ = 1.94: *p* = 0.046), where two kennels saw reductions in the score, three had increases and one had no significant change. 

### 3.2. Trainer Reports 

Due to time constraints, only five of the trainers completed the final interview. All of the trainers reported the regime to be relatively easy, scoring between one and three (on a five-point scale where one is very easy and five very difficult (Table 3), and on average taking two minutes per dog. Three of the five reporting trainers thought their dogs tolerated brushing better at the end, than at the beginning, of the study, and three reported a perceived slight improvement in gum health in those dogs that were brushed weekly and a large improvement on those brushed daily. The other two reported no perceived change in gum health. When considering calculus build up, all five reporting trainers believed that they saw an improvement in the daily group and three also in the weekly group. All of the trainers reported feeding soaked food and all provided roasted bones, but at differing frequencies. 

## 4. Discussion

Brushing by trainers over a two-month period was effective at reducing both calculus and gingivitis in racing dogs, even in a population with variable dental health at the start of the trial. Previous brushing trials have started with a clean mouth, using dental scaling and brushing to achieve this [22,26], and thus this present research is, to our knowledge, the first demonstration of the value of brushing without prior scaling. 

Calculus was, on average, significantly reduced, even when brushing was only weekly. However, when the teeth were brushed daily, the effects were more pronounced. In the case of gingivitis, there was also a reduction for both treatment groups, but only daily brushing was sufficient to lead to a statistically significant reduction. 

These results support findings by other authors, both during controlled laboratory trials, in which brushing daily [22,26], or every other day [24] was deemed necessary to protect gingival health and to reduce plaque and calculus accumulation, and home-based studies, which suggest the importance of daily dental care [20].

When examining individual teeth groups, calculus and gingivitis reduction was only significant on the canines and pre-molars, whilst change on the incisors was minimal. Using the examination here, there does not seem to be a major problem with the incisors, as the initial levels of calculus and gingivitis were relatively low; however, further examination, including probing, would be necessary to confirm tooth health. 

Calculus is thought to contribute to periodontal disease by creating an anaerobic environment and, hence, promoting the development of pathogenic bacteria [12]. Studies have shown that 52% of the variation in periodontal disease severity can be explained by the calculus level [29]; hence, by reducing calculus build up, the brushing intervention is likely to have a significant effect on the progression of periodontal disease.

Here, the calculus was significantly reduced even by weekly brushing. Since there were no other differences in husbandry between the groups, this suggests that not only is accumulation prevented, but counter to popular belief (e.g., [30]), calculus can also be decreased by brushing with enzymatic toothpaste. The mechanism for this is unclear, but it could reduce the adhesion and hardness of the calculus deposits, which are then more easily removed by chewing. However, the average level of calculus was not reduced to zero over the two-month period; hence, for complete removal, either a longer time period, or dental scaling under anaesthetic is likely to be required [30]. What is more, the remaining calculus is most likely on, or near, the gumline, where it facilitates plaque build-up and is of most concern for periodontal disease, and in need of scaling. The longer-term effects of the regime, and the need for a clinical intervention, particularly in more severely affected individuals, are therefore still worthy of investigation. However, the results suggest an overall value in brushing, which, as a lower cost intervention, may encourage financially limited trainers or owners to engage in some dental care. 

Gingivitis is inflammation of the gums, which can be painful. Here, we confirm that gingivitis can be reduced by brushing, even in the absence of dental scaling, but in order to produce a significant change over a two-month period, daily (as compared to weekly) brushing is required. Our findings support those in a laboratory setting, (e.g., [11,22,25]) and demonstrate the efficacy of brushing in an applied setting. When exploring individual teeth groups, the effects on the incisor teeth were non-significant, whilst daily brushing was effective at reducing calculus and gingivitis on the canine and pre-molar teeth. Since these are larger teeth, the differences may be due to trainers paying more attention to them and hence the brushing efficacy was greater. In contrast, the lack of effect on the incisors may be due to their small size and consequent neglect during the brushing regime. However, initial gingivitis and calculus levels tended to be lower on the incisors, possibly due to better saliva coverage of these front teeth, meaning plaque and subsequently calculus does not build up, or because less food makes contact with incisors. Additionally, the low initial levels likely make the chance of a significant reduction lower. 

The oral examination of conscious dogs has some limitations for the accurate measurement of periodontal health [31,32], but it was sufficiently sensitive to detect significant changes in this blinded controlled trial. However, it was not possible to examine the molar teeth, as it was deemed too invasive for an unfamiliar experimenter to adequately manipulate the mouth of non-habituated doliocephalic dogs; hence, we cannot draw any conclusions about the efficacy of brushing for this tooth group. We suggest in future trials, gradual and reward-based habituation to dental examination may also facilitate examination of the molars. In addition, the interaction between kennel establishment and treatment group showed that, for the pre-molar calculus level, brushing was only effective in two kennel establishments, suggesting that trainers varied in the efficacy of their brushing technique or that other factors, such as feeding regime, moderated the effect of brushing.

Since the trainers implementing the routine reported a minimal time commitment and positive experiences, generally believing brushing to be beneficial, we suggest that daily brushing should be recommended for racing greyhounds, and that demonstrations and instructions should include attention to all teeth groups including incisors and pre-molars. It is important to note that the trainers in this trial were given a financial incentive to participate, and hence may not be representative of the general population. However, the trainers were selected on the basis that they did not previously embark upon a regular dental care regime (whilst many of their peers did), and here, over a two-month period, compliance was seen to be relatively high. This suggests that the regime may be practicable for a large proportion of the trainer population. Similar trials now need to be conducted on the retired greyhound population, since this has also been shown to present particularly high levels of periodontal disease.

## 5. Conclusions

Weekly and daily teeth brushing by greyhound care staff over two months, can result in significant reductions in calculus, but to reduce gingivitis daily brushing is required. The effects are less noticeable on the front incisor teeth. Since the staff implementing the routine reported a minimal time commitment and positive experiences, daily brushing is recommended for racing greyhounds, and emphasis should be placed on brushing all teeth groups.

## Figures and Tables

**Figure 1 animals-11-01869-f001:**
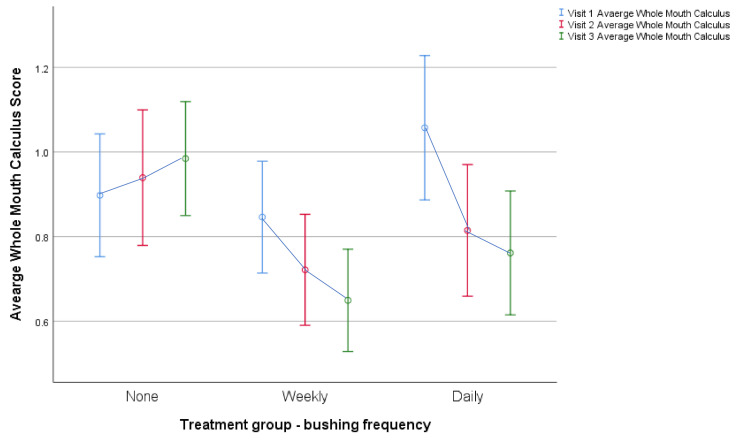
Change in Average Whole Mouth Calculus score between visit 1 (baseline), visit 2 (4 weeks later) and visit 3 (8 weeks later) in each of the three treatment groups (none, weekly and daily brushing). Circles indicate means and lines indicate standard deviations.

**Figure 2 animals-11-01869-f002:**
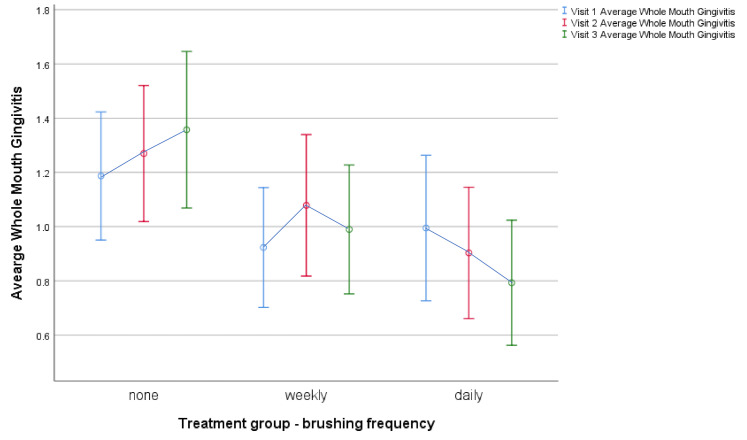
Change in Average Whole Mouth Gingivitis score between visit 1 (baseline), visit 2 (4 weeks later) and visit 3 (8 weeks later) in each of the three treatment groups (none, weekly and daily brushing). Circles indicate means and lines indicate standard deviations.

**Table 1 animals-11-01869-t001:** Four-point scale used to rate the level of calculus.

Score	Definition
0	No observable calculus
1	Less than 25% coverage of calculus
2	Between 25–<75% coverage of calculus
3	75–100% coverage of calculus

**Table 2 animals-11-01869-t002:** Four-point scale used to rate the level of gingivitis.

Score	Definition
0	No inflammation
1	Mild inflammation, slight change in colour, little change in texture of any portion of the gingival unit
2	Moderate inflammation (moderate glazing, redness, oedema and/or hypertrophy) of the gingival unit
3	Severe inflammation (marked by redness and oedema/hypertrophy, spontaneous bleeding or ulceration) of the gingival unit

**Table 3 animals-11-01869-t003:** Number of dogs, feeding and bone routine at each kennel establishment and perceptions of the trainer/carer regarding the dog’s behaviour and dental health pre- and post-trial.

Variable Name	Trainer 1	Trainer 2	Trainer 3	Trainer 4	Trainer 5	Trainer 6
Number of dogs completing the trial in each kennel	27	26	37	18	25	27
Perceived ease of brushing (1 very easy–5 very difficult)	1	2	1	1–2	3	1
Perceived change in the average dog in the kennels’ reaction to brushing over 2 months (+ slight improvement: 1 point on 5 point scale; ++ major improvement: 2 points on 5 point scale)	None	++	++	+	None	Failed to complete interview
Reported average change in gum health with weekly brushing (+ slight improvement: 1 point on 5 point scale; ++ major improvement: 2 points on 5 point scale)	+	+	+	None	None	NA
Perceived change in gum health with daily brushing (+ slight improvement: 1 point on 5 point scale; ++ major improvement: 2 points on 5 point scale)	++	++	++	None	None	NA
Perceived change in calculus with weekly brushing	++	+	+	++	+	NA
Perceived change in calculus with daily brushing on a five point scale (+ slight improvement: 1 point on 5 point scale; ++ major improvement: 2 points on 5 point scale)	++	++	++	+	+	NA
Opinion of what most contributed to improvement, product (P) or brushing (B)	P	P and B	P	Neither	P and B	NA
Reported time spent brushing each dog’s teeth	3–4 min	1 min	2 min	1–2 min	1.5 min	NA
Initial % of dogs with bleeding gums	30	25	50	5	25	NA
Ultimate % of dosg with bleeding gums	0	5	0	0	0	NA
Food soaked in water	Unknown	Cold water 60 min	Hot water 5–10 min	Warm water 3 h	Hot water 60 min	NA
Bone feeding regime	Reported roasted whenever in magnetic therapy box	Reported roasted whenever in magnetic therapy box	Roasted every 3–4 weeks	Roasted monthly	Roasted every 1–2 weeks	NA

**Table 4 animals-11-01869-t004:** Results of a GLM testing the effects of baseline measure, treatment group (control (none), weekly or daily brushing) and kennel establishment on each measure of dental health after two months.

Variable	Dog’s Baseline		Treatment		Kennel Establishment		Comparisons					
Variable Name	F	*p*	F	*p*	F	*p*	Control-Weekly	*p*	Control-Daily	*p*	Weekly-Daily	*p*
Average whole mouth calculus	27.13	<0.001	10.76	<0.001	8.64	<0.001	C > W	<0.001	C > D	<0.001	W > D	NS
Incisor calculus	33.60	<0.001	1.18	NS	0.62	NS	-	-	-	-	-	-
Canine calculus	11.04	0.001	5.36	0.006	6.18	<0.001	C > W	0.003	C > D	0.018	W > D	NS
Pre-molar calculus	15.21	<0.001	8.87	<0.001	9.99	<0.001	C > W	<0.001	C > D	0.001	W > D	NS
Average whole mouth gingivitis	25.21	<0.001	4.57	0.012	3.51	0.005	C > W	NS	C>D	0.003	W > D	NS
Incisor gingivitis	21.31	<0.001	2.56	NS	1.22	NS	-	-	-	-	-	-
Canine gingivitis	14.44	<0.001	3.92	0.022	3.79	0.003	C > W	NS	C > D	0.006	W > D	NS
Pre-molar gingivitis	14.84	<0.001	5.46	0.005	3.54	0.005	C > W	NS	C > D	0.002	W > D	NS

**Table 5 animals-11-01869-t005:** Comparison of calculus and gingivitis scores (whole mouth average and individual teeth groups) between visits 1 and 3 in each of the treatment groups. All values are mean± standard deviation.

Variable	Control		Weekly		Daily	
Variable Name	Pre (Visit 1)	Post (Visit 3)	Pre (Visit 1)	Post (Visit 3)	Pre (Visit 1)	Post (Visit 3)
Whole Mouth Calculus	0.90 ± 0.496	0.99 ± 0.460	0.91 ± 0.478	0.70 ± 0.471	1.06 ± 0.534	0.79 ± 0.463
Incisor Calculus	0.04 ± 0.153	0.09 ± 0.211	0.10 ± 0.281	0.09 ± 0.314	0.13 ± 0.347	0.09 ± 0.259
Canine Calculus	1.11 ± 0.637	1.09 ± 0.641	1.20 ± 0.634	0.82 ± 0.654	1.26 ± 0.671	0.87 ± 0.513
Pre-molar calculus	1.57 ± 1.061	1.80 ± 0.876	1.42 ± 0.948	1.19 ± 0.852	1.81 ± 0.954	1.42 ± 0.931
Whole Mouth Gingivitis	1.19 ± 0.804	1.37 ± 0.990	1.03 ± 0.839	1.01 ± 0.802	1.01 ± 0.846	0.87 ± 0.771
Incisor Gingivitis	0.73 ± 0.904	0.98 ± 1.198	0.56 ± 0.899	0.62 ± 0.885	0.57 ± 1.000	0.54 ± 0.884
Canine Gingivitis	1.56 ± 0.826	1.65 ± 0.900	1.41 ± 0.909	1.36 ± 0.865	1.36 ± 0.802	1.16 ± 0.813
Pre-molar Gingivitis	1.33 ± 0.931	1.47 ± 1.005	1.13 ± 0.956	1.06 ± 0.858	1.10 ± 0.954	0.90 ± 0.843

## Data Availability

Data available on request.

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
