# Peer review of "Weekly and Daily Tooth Brushing by Care Staff Reduces Gingivitis and Calculus in Racing Greyhounds"

_animals, 2021, doi:10.3390/ani11071869_

Round 1

Reviewer 1 Report

Comments for Authors:

The topic addressed sounds very interesting. The introduction clearly describes what research has been done so far in the area of the analyzed topic. In addition, the purpose of the current study was specified. The material and methods section has been described in details. The research group (160 dogs) also deserves attention. The results were clearly presented. In my opinion, this manuscript can be recommended for publication.

Author Response

Many thanks for your kind comments.

Reviewer 2 Report

This article describes "Weekly and daily tooth brushing by care staff reduces gingivitis and calculus in racing greyhounds". I think the results in this manuscript are of interest, particularly in terms of the data demonstrating the beneficial effects of the dog's oral hygiene. The data are sound. However, there are few concerns that should be addressed.

1. It seems that heading number 3 is missing. Please add the heading number and correct.

2. How did you decide on the sample size? Please describe how to decide.

3. Calculus cannot be removed by brushing alone. The decrease in calculus on line 312 may have been removed by roasted bone.

Author Response

May thanks fo ryour comments below. In  answer 

  1. It seems that heading number 3 is missing. Please add the heading number and correct. - we have corrected the numbering throughout 

2. How did you decide on the sample size? Please describe how to decide.

To decide on the size of our study we were constrained in part by the number of greyhound trainers that we could recruit and travel to in the time period of the study. We were aiming to be able to detect medium to large effect sizes across several measures (and between 3 groups). For simplicity we looked at power calculations for a 2-sample t test (to compare just a pair of groups) while noting our actual design has 3 groups and we control for baseline measures in the model and between kennel differences in the design. Here to detect a medium effect size (0.5) with power 0.8 while assuming equal variability in the measure between groups would require a sample size of 64 per group whilst for a large effect size (0.8) would require a sample size of 26 per group. Our actual samples sizes of 52-54 dogs per group is somewhere between these figures (being able to detect effects of roughly size 0.55 - 0.6) which when considering in the added complexity of the model seemed an appropriate sample size. 

3. Calculus cannot be removed by brushing alone. The decrease in calculus on line 312 may have been removed by roasted bone.

In order to add further clarity we have added to line 312, "Here, the calculus was significantly reduced even by weekly brushing. Since there were no differences in husbandry between the groups, this suggests that not only is accumulation prevented, but counter to popular belief (e.g. White 1991), calculus can also be decreased by brushing with enzymatic toothpaste. The mechanism for this is unclear, but it could reduce  adhesion and hardness of the calculus deposits, which are then  more easily removed by chewing.

Reviewer 3 Report

I consider this manuscript an excellent contribution to the field of Veterinary Dentistry.

The study is extremely well designed and the results answer a very relevant question in the clinical practice of companion animals.

Apart from minor typos and text and table formatting, I don't identify major errors. Just these aspects could be clarified and corrected by the authors:

Line 19. A comma is missing in “The effects however were not noticeable on the front incisor teeth Since staff..)”.

Line 71: Instead of use the word “hardens”, it is recommended “mineralizes”.

Line 176: The chapters “2.4. Brushing demonstration” and “2.5. Instructions” could be combined and adapted in one chapter called about the brushing protocol.

Author Response

We have made all the changes detailed below and proof read the MS to correct any typos or formatting issues.